

# Movements and use of space by Mangrove Cuckoos (*Coccyzus minor*) in Florida, USA

John David Lloyd

Vermont Center for Ecostudies, Norwich, VT, United States of America
Ecostudies Institute, East Olympia, WA, United States of America

## ABSTRACT

I used radio-telemetry to track the movements of Mangrove Cuckoos (*Coccyzus minor*) captured in southwest Florida. Relatively little is known about the natural history of Mangrove Cuckoos, and my goal was to provide an initial description of how individuals use space, with a focus on the size and placement of home ranges. I captured and affixed VHF radio-transmitters to 32 individuals between 2012 and 2015, and obtained a sufficient number of relocations from 16 of them to estimate home-range boundaries and describe patterns of movement. Home-range area varied widely among individuals, but in general was roughly four times larger than expected based on the body size of Mangrove Cuckoos. The median core area (50% isopleth) of a home range was 42 ha (range: 9–91 ha), and the median overall home range (90% isopleth) was 128 ha (range: 28–319 ha). The median distance between estimated locations recorded on subsequent days was 298 m (95% CI [187 m–409 m]), but variation within and among individuals was substantial, and it was not uncommon to relocate individuals >1 km from their location on the previous day. Site fidelity by individual birds was low; although Mangrove Cuckoos were present year-round within the study area, I did not observe any individuals that remained on a single home range throughout the year. Although individual birds showed no evidence of avoiding anthropogenic edges, they did not incorporate developed areas into their daily movements and home ranges consisted almost entirely of mangrove forest. The persistence of the species in the study area depended on a network of conserved lands–mostly public, but some privately conserved land as well–because large patches of mangrove forest did not occur on tracts left unprotected from development.

# INTRODUCTION

Understanding how animals use space and move through the environment around them can provide important insights into their ecology and conservation (*Kramer & Chapman, 1999*; *Wiens, 2008*; *Holland et al., 2009*). Information concerning an animal's home range—that is, the area in which an organism carries out the day-to-day activities of life (*Burt, 1943*)— can be particularly useful, helping to identify habitat requirements, predict sensitivity to habitat loss and fragmentation, and delineate areas important for conservation. I

Corresponding author
John David Lloyd,
jlloyd@vtecostudies.org,
5355693@gmail.com

documented patterns of movement and described characteristics of Mangrove Cuckoo (*Coccyzus minor* Gmelin) home ranges in southwest Florida, USA. Mangrove Cuckoos are widespread and relatively common in a variety of forested environments throughout the Caribbean and Middle America (*Lloyd, 2013*). In Florida, the northern limit of their geographic distribution, they are uncommon and apparently restricted largely to mangrove forests (*Lloyd, 2013*; *Lloyd & Slater, 2014*). Although the species is of Least Concern globally (*BirdLife International, 2012*), Mangrove Cuckoos in the United States are a high priority for conservation action (*Partners in Flight Science Committee, 2012*) and are considered at risk of becoming threatened (*US Fish and Wildlife Service, 2008*), with some evidence of recent declines in parts of Florida (*Lloyd & Doyle, 2011*). An important obstacle to planning conservation action, however, is the lack of information on the natural history of Mangrove Cuckoos; they remain one of North America's least-studied birds (*Hughes, 2012*).

My goal was to enhance understanding of the natural history of Mangrove Cuckoos by providing an initial description of space use; as with other facets of the species' ecology, basic patterns of space use are undocumented. To address this information gap, I sought to quantify patterns of movement among individuals, estimate the amount of area required to support a Mangrove Cuckoo home range, and describe qualitatively the land-cover types in which Mangrove Cuckoos will establish a home range. Information on area requirements and habitat use may help inform future conservation efforts. I did not document what sorts of activities birds engaged in during the period of time that I followed them (e.g., whether they were nesting), so here I adopt a simple empirical approach of allowing the movement of individual birds to define an area of concentrated use that I refer to as a home range (sensu *Burt, 1943*).

## METHODS

### Study area

I captured Mangrove Cuckoos from 2012 to 2015 at JN "Ding" Darling National Wildlife Refuge (26.44°N, −82.11°W) (hereafter, "Ding Darling NWR") on the barrier island of Sanibel and at San Carlos Bay—Bunche Beach Preserve (26.48°N, −81.97°W) on the nearby mainland coast in Fort Myers. The study area, however, encompassed all of the locations where I relocated marked birds, ranging from near Port Charlotte to Fort Myers Beach (Fig. 1). Mangrove forests fringe protected coastlines in this area and are dominated by red (*Rhizophora mangle* L.) and black (*Avicennia germinans* L.) mangrove, with lesser numbers of white mangrove (*Laguncularia racemosa* C. F. Gaertn.). Canopy height in general was less than 7 m. The inland edge of most mangrove forest in the region abuts developed land, where nearly all uplands have been cleared of native vegetation for commercial and residential development. Where uplands have been protected—almost exclusively on Sanibel—adjacent forest types include hammock forests dominated by southern live oak (*Quercus virginiana* Mill.) and a variety of tropical hardwoods, savannas of cabbage palm (*Sabal palmetto* Lodd. ex Schult.f.), and pure stands of buttonwood (*Conocarpus erectus* L.) (*Cooley, 1955*).

The climate of the area is tropical (*Duever et al., 1994*). Air temperatures remain relatively warm throughout the year, with mean monthly temperature ranging from 17.8 °C in
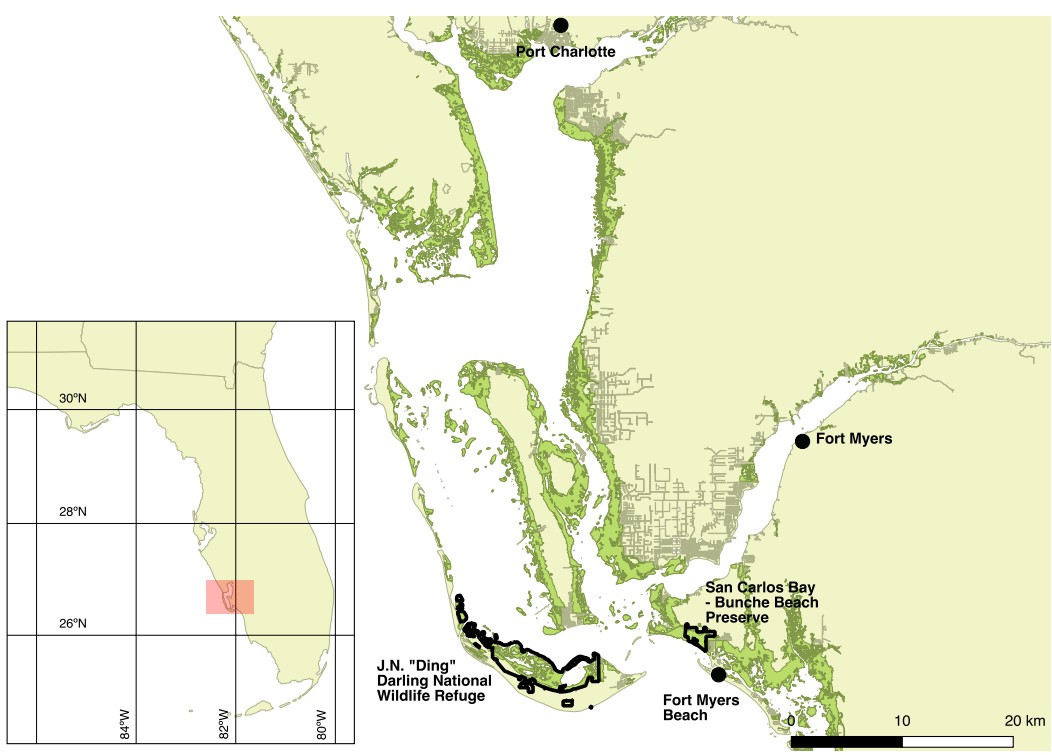

**Figure 1** **Map of the study area.** Study area (red shaded box on the inset map) in southwest Florida, USA, where Mangrove Cuckoos (*Coccyzus minor*) were radio-tracked during 2012–2015. Individuals were captured in mangrove forest (green shading) within two protected areas: JN "Ding" Darling National Wildlife Refuge, located on the barrier island of Sanibel, and San Carlos Bay—Bunche Beach Preserve, located on the mainland in the city of Fort Myers. Individuals were tracked as far north as Port Charlotte, and as far south as Fort Myers Beach.

January to 28.1 °C in August (based on climate data from 1892 to 2012 collected in Fort Myers; available online at http://www.sercc.com). Frosts are uncommon, especially in mangroves. Most (65%) of the mean annual precipitation (136 cm) falls during convective storms in the pronounced wet season (June–September). Weather between October and May is drier and cooler, and precipitation that falls during the dry season is generally driven by the passage of cold fronts. Tropical cyclones strike occasionally, although none affected the area during this study.

## Field methods

I located birds by broadcasting a recorded vocalization of Mangrove Cuckoo, to which individuals respond readily when present (*Frieze, Mullin & Lloyd, 2012*), in areas of suitable habitat (mangrove forest) that could be accessed by boat, on foot, or by motor vehicle. In 2012, searches were conducted between March and August; in 2013, between February and August; and then continually from February 2014 to June 2015. The start and end dates of searches in 2013 and 2014 were dependent on the availability of personnel to assist with searches.

The vocalization used during playback (*Hardy, 1998*) was downloaded from the website of the Florida Museum (http://www.flmnh.ufl.edu/birds/florida-bird-sounds/)

and consisted of the typical guttural series of "cah" notes, lasting for 8 s. Broadcasts were made using a small handheld speaker and an MP3 player, with the volume set to a level at which the sound could be distinguished by a human observer at a distance of approximately 100 m. I listened quietly after each playback, repeating the broadcast up to 3 times if no individuals were detected.

Once a bird had been located, it was lured into a mist net via playback of recorded vocalizations. Upon capture, each bird was marked with an aluminum US Fish and Wildlife Service leg-band and a unique combination of three colored plastic leg-bands. A VHF radio-transmitter (American Wildlife Enterprises, Monticello, Florida and ATS, Isanti, Minnesota) was attached using flat, 2.5-mm-wide elastic fabric to create leg loops as per *Rappole & Tipton (1991)*. The transmitter and harness collectively weighed 1.8 g, or approximately 2.9% of the average mass of Mangrove Cuckoos captured in this study (mean body mass = 62.5 g; $n = 46$). Protocols and materials used in capture, handling, and marking were designed in accordance with guidelines presented by *Fair, Paul & Jones (2010)*. This research was conducted with the permission of the US Fish and Wildlife Service (Special Use Permit No. 13036), the USGS Bird Banding Laboratory (Bird-Banding Permit No. 23726 issued to JDL), and the State of Florida (Scientific Collecting Permit No. LSSC-11-00048A).

Birds were released as soon as possible after capture (average time between capture in the mist net and release of a radio-marked bird was 27 min). I attempted to relocate radio-marked birds every 1–3 days using a handheld antenna, although this frequency of relocation was possible only for birds that remained in the core of the study area. Individuals that moved long distances or occupied remote areas that could only be searched by plane were relocated less frequently, generally every 2–3 weeks.

When an individual could not be located after multiple ground-based searches, a fixed-wing airplane was used to search a wider area. Aerial searches typically focused on an area within 60 km of the last known location. Location of individuals detected during aerial searches was estimated from the plane's Global Positioning System (GPS) after the signal had been localized using directional antennae and close circling by the pilot.

Radio-marked individuals were tracked throughout each field season (see above for dates) or until multiple aerial searches failed to detect them. The nominal battery life of the transmitters ranged from 3 to 6 months depending on the unit, but in general I could not distinguish battery failure from permanent emigration out of the search area.

## Estimating telemetry error

To test the telemetry system, a naïve observer used biangulation to identify the location of a radio transmitter that had been placed in a known location by a second observer. The transmitters were placed on horizontal limbs of mangrove trees in locations that were representative of perches used by Mangrove Cuckoos. I conducted 16 trials; six in February of 2012 and 10 in July of 2012. The same observer was used in every trial. In 14 trials, the observer was able to obtain bearings from land, but in the other two trials the location of the hidden transmitter required the observer to take bearings from a kayak. I calculated

error as the distance between the actual location of the transmitter as determined by a handheld GPS unit and the location estimated from biangulation.

## Efficacy of aerial searches

I also conducted a test of the efficacy of aerial searches from a fixed-wing airplane. On a single day, a pilot flew at different altitudes above a transmitter positioned at a known location in a mangrove forest. The plane passed directly over the transmitter at 305 m, 457 m, and 610 m, and then flew passes at different distances to either side of the transmitter, again repeating passes at each of the three altitudes.

## Statistical analysis of movements and space use

I estimated the location of marked birds by triangulating the signal based on compass bearings and GPS locations obtained in the field. I described home ranges of radio-marked Mangrove Cuckoos using the Brownian bridges movement model of *Horne et al. (2007)*, as implemented in the R package adehabitatHR (*Calenge, 2006*). This model requires time-stamped locations and two smoothing parameters, one related to the speed at which the organism moves through space (the Brownian motion variance parameter) and one that describes the imprecision of estimated locations. I calculated the Brownian motion variance parameter using the likelihood method proposed by *Horne et al. (2007)* and implemented by the liker function in the adehabitatHR package. I used the results of the ground-based telemetry-error tests to calculate the standard deviation of the mean location error, the second smoothing parameter (I have only qualitative information about error during aerial searches). In estimating the boundaries of home ranges, I censored from analysis any individuals with ≤20 relocations due to concerns about small-sample bias. Based on the recommendation of *Borger et al. (2006)*, I defined the total home range as the 90% isopleth of the utilization distribution, and the core home range as the 50% isopleth. Location data used to estimate the home-range boundaries are available in *Lloyd (2017)*.

Home-range boundaries for Mangrove Cuckoos in this area tended to include large areas of open water, which I did not include in calculations of home-range area. The amount of open water within each home range was calculated using a shapefile of the Florida coastline (version 2004) published by the State of Florida (available at http://www.fgdl.org) and then subtracted from the area within the 90% and 50% isopleths. Home-range size calculations were performed within QGIS version 2.16.3 (*QGIS Development Team, 2016*); all other analyses were conducted in R 3.2.4 (*R Core Team, 2016*).

I used the shapefile (version April 2015) published by the Fish and Wildlife Research Institute (FWRI) at the Florida Fish and Wildlife Conservation Commission to determine the distribution of mangrove vegetation within the study area (available at http://www.fgdl.org). I determined protected area boundaries using version 1.4 of the US Geological Survey's Protected Areas Database of the United States (available at: http://gapanalysis.usgs.gov/padus/).

## RESULTS

### Telemetry error

The estimated mean telemetry error associated with ground-based searches was 35.1 m (SD = 28.6 m; range = 5.7 m–105.3 m).

### Efficacy of aerial searches

Flying directly over the transmitter at 305 m altitude, the signal was detected 1.1 km before the plane passed over the transmitter and was lost when the plane had passed 1.0 km beyond the location of the signal. At this altitude, the signal was not detected at the 1 or 2 km offset passes. At 457 m altitude, the signal was detected 1.8 km before the plane passed over the transmitter and was lost when the plane had passed 800 m beyond the transmitter. The signal was located on offset passes as far as 2 km adjacent to the path directly over the signal. At 610 m altitude, the signal was detected 1.7 km before the plane passed over the transmitter and was lost when the plane had passed 900 m beyond the transmitter. The signal was located on offset passes as far as 2 km adjacent to the path directly over the signal. These results suggest that, at altitudes typical of those maintained during aerial searches (>400 m), the detection radius for a transmitter on the ground was approximately 1–2 km. Patches of mangrove forests in the study area were always <4 km in width, and most were <1 km wide (e.g., Fig. 1).

### Movements and space use by Mangrove Cuckoos

I captured 46 individuals between 2012 and 2015. I did not recapture or resight any marked individuals outside of the year in which they were initially captured (except for one individual captured in late 2014 and tracked into early 2015). I captured individuals in every month except February, but most captures ($n = 27$) occurred between March and May (Fig. 2). I radio-marked 32 of these individuals, and obtained an adequate number of relocations for 16 of these to describe a home range. Of the 16 individuals censored from the home-range analysis due to small sample size, six were tracked for relatively long periods of time (127, 123, 114, 111, 103, and 45 days, respectively) but occupied areas where transmitter signals could only be detected by plane and thus were relocated infrequently. The other 10 were transient (or carried transmitters that failed prematurely); most of these individuals were known to be present in the study area for <2 weeks (average number of days known present = 13; range = 2–31 days).

In general, individuals moved widely from day to day. The median distance between estimated locations recorded on subsequent days was 298 m (95% CI [187 m–409 m]), but variation within and among individuals was substantial, and individuals were occasionally found >1 km from their location on the previous day (Fig. 3). Notable movements included a flight taken by individual 150.919 from its home range in Ding Darling NWR to the San Carlos Bay—Bunche Beach Preserve and back again, a round-trip distance of roughly 35 km. This individual was located on its home range at 07:01 on 18 July 2012, but by the following morning at 09:59 it had moved to a location in San Carlos Bay—Bunche Beach Preserve on the mainland, a straight-line distance of 16.8 km. It was not located on 20 July. On 21 July at 08:06 it had returned to nearly the same location where it had been found

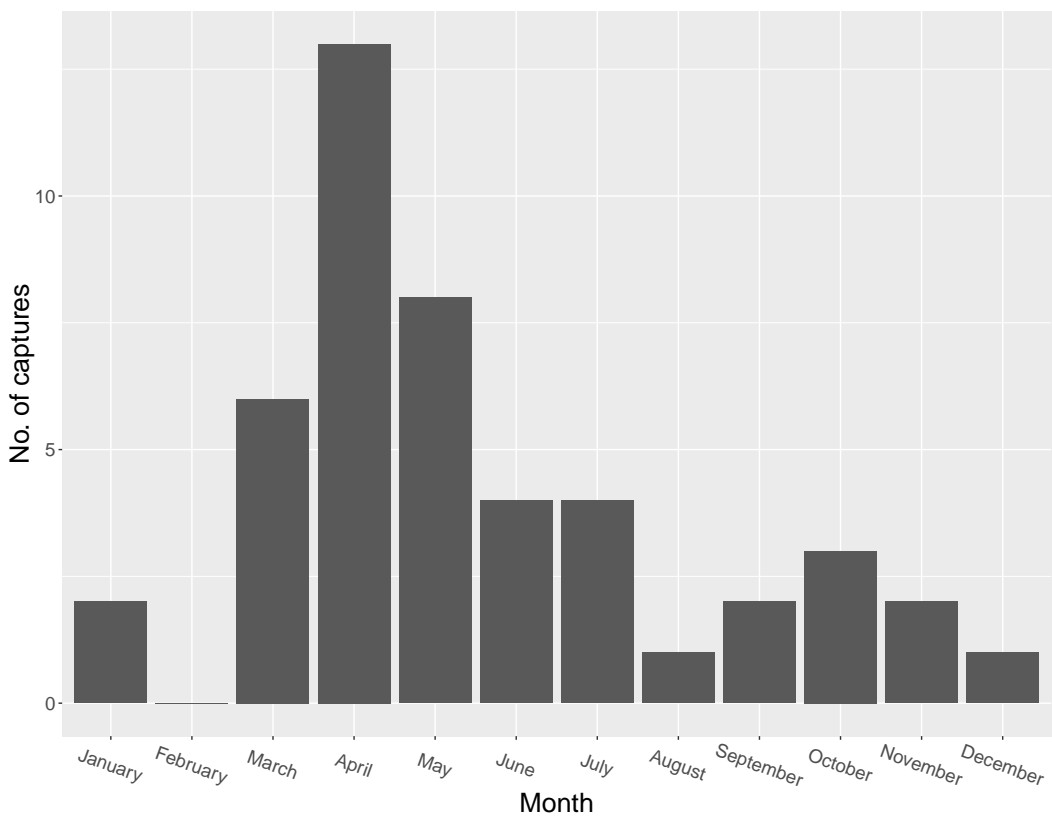

**Figure 2  Seasonal distribution of captures of Mangrove Cuckoos (*Coccyzus minor*) (*n* = 46) in southwest Florida during 2012–2015.**

on 18 July. This individual then remained on its home range on Sanibel until at least 21 November 2012, and during that time made no other similar movements. Although the purpose of that single long-distance movement is unknown, it was evidently not part of a dispersal event to a new home range.

Home-range area was generally large but variable among individuals (Table 1). Home-range area did not covary with the length of the period during which I tracked each individual (total home range: $r = 0.30$, 95% CI [−0.23–0.69]; core area: $r = 0.26$, 95% CI [−0.25–0.66]) or with the number of times an individual was relocated (total home range: $r = 0.29$, 95% CI = −0.24–0.69; core area: $r = 0.16$, 95% CI [−0.35–0.59]). Of the 16 individuals for which I estimated a home range, 11 were last detected within its boundaries. The other 5 individuals (150.613, 150.757, 149.881, 148.872, and 149.281) were later located 1–3 times at locations far removed from the home-range boundaries (c.a. 12–55 km from the last estimated location within the home range). None of these five individuals ever returned, and thus presumably had abandoned the home range and were in the process of dispersing when last located. Timing of departure, for these five individuals, ranged from early May (149.281) to late July (150.757). The trigger for these dispersal events is unknown.

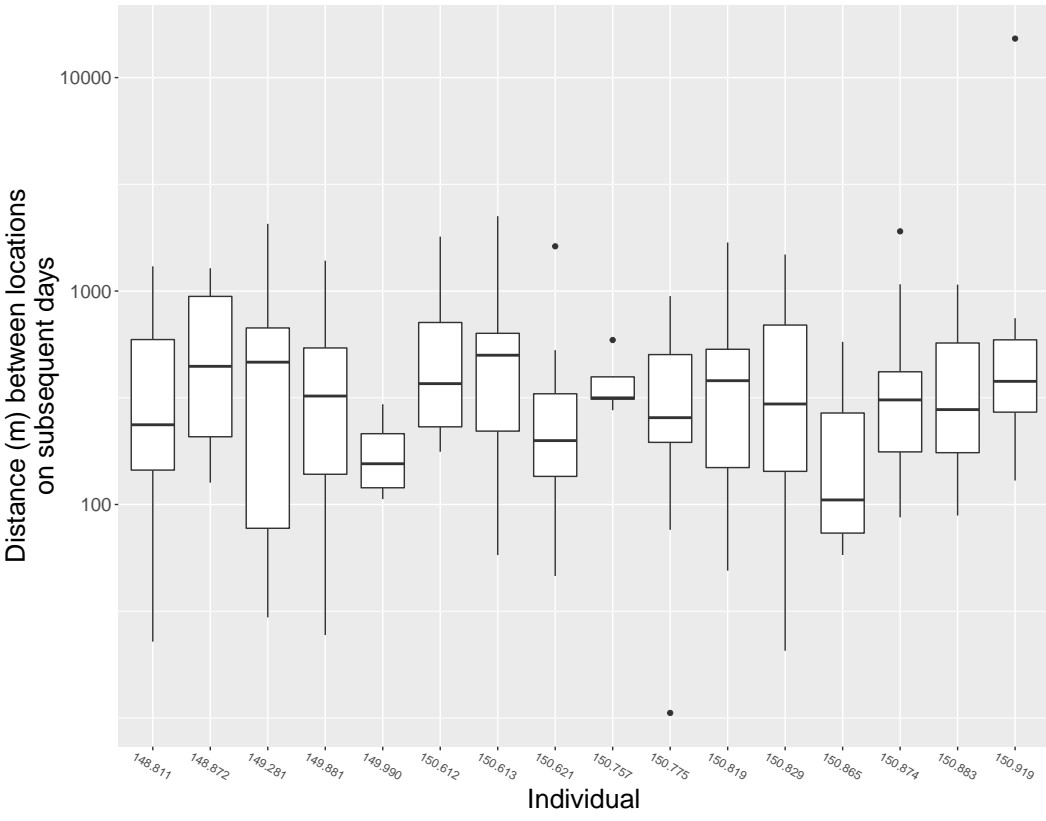

**Figure 3** **Daily movement distances of Mangrove Cuckoos.** Distance between estimated locations of individual radio-tagged Mangrove Cuckoos (*Coccyzus minor*) on subsequent days (i.e., estimated locations taken 18–28 h apart) in southwest Florida from 2012 to 2015. Only individuals ($n = 16$) with an adequate number of relocations to estimate home-range boundaries are included.

The same areas were frequently used as home ranges by different birds in different years, but concurrent use of overlapping home ranges or core-use areas was observed in only one instance. Three individuals—150.775, 150.829, and 150.819—occupied broadly overlapping (i.e., >50% overlap) home ranges and core-use areas at the same time in San Carlos Bay—Bunche Beach Preserve. I did not observe interactions among these individuals, so it is unclear whether they were part of a social unit. However, all three individuals were located in close proximity to one another on numerous occasions throughout the period during which they were tracked.

Nearly 75% of estimated locations of marked Mangrove Cuckoos fell within areas classified as mangroves (756 locations from a total of 1,015 locations gathered during the course of the study) and 94% of all estimated locations fell within 100 m of mangrove vegetation as defined by the FWRI shapefile. Mangrove vegetation in the study area is limited primarily to protected areas, and as consequence nearly every (99%; $n = 1,002$ locations) estimated location of a Mangrove Cuckoo occurred within a protected area. In addition to the two main capture areas, Ding Darling NWR ($n = 590$ locations) and San Carlos Bay—Bunche Beach Preserve ($n = 156$ locations), other protected areas used by Mangrove Cuckoos included conservation lands managed by Sanibel-Captiva Conservation

**Table 1 Home-range characteristics of Mangrove Cuckoos.** Home-range characteristics of 16 Mangrove Cuckoos (*Coccyzus minor*) tracked via radio-telemetry on the southwest coast of Florida from 2012 to 2015.

| Individual | N | Home-range area (ha) | | Tracking dates |
|---|---|---|---|---|
| | | Core area[a] | Total[b] | |
| 148.811 | 57 | 42 | 153 | 3 Mar–12 Jun 2014 |
| 148.872 | 39 | 79 | 243 | 11 Mar–27 May 2014 |
| 149.281 | 20 | 91 | 243 | 4 Apr–6 May 2014 |
| 149.881 | 47 | 70 | 294 | 18 Apr–27 Jun 2014 |
| 149.990 | 26 | 9 | 28 | 25 Nov 2014–18 Jan 2015 |
| 150.612 | 37 | 24 | 92 | 28 Apr–16 Jun 2012 |
| 150.613 | 53 | 15 | 104 | 7 Jun–22 Aug 2013 |
| 150.621 | 42 | 30 | 107 | 8 May–4 July 2012 |
| 150.757 | 31 | 64 | NA | 9 May–30 Jul 2013 |
| 150.775 | 70 | 28 | 125 | 14 May–22 Aug 2013 |
| 150.819 | 42 | 60 | 201 | 18 Jun–22 Aug 2013 |
| 150.829 | 36 | 42 | 132 | 9 Jul–22 Aug 2013 |
| 150.865 | 58 | 9 | 36 | 20 May–22 Aug 2013 |
| 150.874 | 76 | 76 | 319 | 15 Mar–15 Jul 2013 |
| 150.883 | 91 | 65 | 164 | 16 Mar–22 Aug 2013 |
| 150.919 | 20 | 24 | 86 | 8 Jul–10 Aug 2012 |
| Mean | | 45.5 (SD = 26.8) | 155.1 (SD = 88.3) | |
| Median | | 42 | 132 | |

**Notes.**
[a] 50% isopleth from a Brownian bridges analysis.
[b] 90% isopleth from a Brownian bridges analysis.

Foundation ($n = 68$), Charlotte Harbor Preserve State Park ($n = 35$), Estero Bay Preserve State Park ($n = 22$), and Matlacha Pass NWR ($n = 6$).

## DISCUSSION

Home-range size of Mangrove Cuckoos captured on public land in southwest Florida was substantially larger than predicted based on the allometry of space use by animals (*Schoener, 1968*; *Mace & Harvey, 1983*). Indeed, with a median home-range size of 132 ha, space use by Mangrove Cuckoos is similar to that of a small raptor such as Red-shouldered Hawk (*Buteo lineatus* Gmelin; average home-range size = 135 ha) (*Peery, 2000*), even though its body size is roughly 15% that of the Red-shouldered Hawk. Little information exists on home-range size of other New World cuckoos. Yellow-billed Cuckoos (*Coccyzus americanus* Linnaeus) in riparian forests in Arizona occupied home ranges that averaged 39 ha (95% kernel-density estimate) to 51 ha (minimum convex polygon) during the breeding season (*Halterman, 2009*), and a single Banded Ground-cuckoo (*Neomorphus radiolosus* Sclater & Salvin)—a distantly related and far larger species—occupied a home-range in Ecuador estimated to consist of 42.2 ha (MCP) to 49.9 ha (95% kernel-density estimate) (*Karubian & Carrasco, 2008*). Likewise, information on space use by other birds of mangrove forest is
scarce; Yellow-billed Cotinga (*Carpodectes antoniae* Ridgway), a substantially larger (85–90 g) inhabitant of mangrove forests in Costa Rica and Panama, used somewhat smaller home ranges (31.2 ha and 107.2 ha, respectively, during the breeding and non-breeding seasons) and core-use areas (6.6 ha and 24.3 ha, respectively) (*Leavelle et al., 2015*).

The Mangrove Cuckoos tracked in this study showed no inter-annual site fidelity. I documented several instances in which the same patch of mangrove was occupied by a different individual in each year of the study. Indeed, during the course of the study, I never recaptured—and only once resighted—an individual marked in a previous year; this suggests a nomadic lifestyle, as has been argued for other *Coccyzus* cuckoos. Although Mangrove Cuckoos were present in the study area year-round, I found no evidence that any individual remained resident in the same area throughout the year.

Why might Mangrove Cuckoos use disproportionately large home ranges and show an apparent tendency to wander widely? Perhaps it is worth considering use of space within the context of the unusual suite of life-history traits that seem to characterize Mangrove Cuckoo and two of its more well-studied congeners: Yellow-billed Cuckoo and Black-billed Cuckoo (*C. erythropthalmus* Wilson). Based on what is known of these species, in addition to occupying large home ranges, they exhibit remarkably rapid developmental rates, are facultative intraspecific brood parasites, have low inter-annual fidelity to breeding sites and highly variable investment in reproduction, and seem to engage in inexplicable, long-distance movements before and after breeding (*Fleischer, Murphy & Hunt, 1985*; *Hughes, 2001*; *Hughes, 2012*; *Hughes, 2015*; *Dearborn et al., 2009*; *Sechrist et al., 2012*). These traits have been explained as an adaptation to a lifestyle centered around exploiting super-abundant but patchy, ephemeral, and unpredictable food resources (*Hamilton III & Hamilton, 1965*; *Nolan Jr & Thompson, 1975*; *Sealy, 1985*; *Barber, Marquis & Tori, 2008*). Evidence for this hypothesis is largely circumstantial, however (e.g., see *Hughes, 1997* for a critique), and it is not clear if the food resources used by Mangrove Cuckoos are as variable as those considered critical for Yellow-billed and Black-billed cuckoos. The diet of Mangrove Cuckoos is known poorly but seems to include a predilection for large invertebrates and small vertebrates (*Lloyd, 2013*) and thus the large home ranges that I observed may have reflected a diet focused on relatively large prey items—a characteristic associated with large home ranges (*Schoener, 1968*)—rather than a diet based on highly variable prey populations. However, as with other *Coccyzus* cuckoos, rigorous tests of these ideas await longer-term studies of breeding biology and natural history. For Mangrove Cuckoos, this would include research that links movement patterns to breeding behavior; tracks individuals across longer temporal and larger spatial scales; and rigorously quantifies diets of adults, juveniles, and nestlings. Finally, I note that I only sampled individuals living in relatively small patches of forest surrounded by urban development. It is possible that birds living in areas with more extensive mangrove forest, for example in Everglades National Park, may exhibit different movement patterns. Comparative research in other areas would help verify the findings presented here.

Although many puzzles remain concerning the natural history of Mangrove Cuckoos, the conditions needed to conserve the species are clear: a network of intact, protected patches of mangrove forest. In south Florida, this network consists almost entirely of

publically owned land. Stands of mangrove forest large enough to support Mangrove Cuckoos do not occur on private land. Some important protected areas—Ding Darling NWR, for example—were established to conserve habitat for wildlife, but other important protected areas, like Charlotte Harbor Preserve State Park, were established largely for shoreline protection and water-quality improvement. No matter what the rationale for investing in mangrove protection, the continued persistence of Mangrove Cuckoos in Florida depends on the preservation of remaining mangrove forests.

## ACKNOWLEDGEMENTS

I gratefully acknowledge the assistance of Rachel Frieze and the staff of the JN ''Ding'' Darling National Wildlife Refuge, especially J Palmer, P Tritaik, T Wertz and J Conrad. I also acknowledge the helpful support and assistance of S Mullin, J and A Kirk, B Fischer, A Parker, K Ramos, M Gallagher, and M McLaughlin. J Hill and C Rimmer provided valuable comments on an earlier version of this manuscript.

### Funding

Funding was provided by the US Fish and Wildlife Service (Cooperative Agreement 401818J604) and the Disney Wildlife Conservation Fund. The funders had no role in study design, data collection and analysis, decision to publish, or preparation of the manuscript.

### Grant Disclosures

The following grant information was disclosed by the author:
US Fish and Wildlife Service: 401818J604.
Disney Wildlife Conservation Fund.

### Competing Interests

The author declares there are no competing interests.

### Author Contributions

- John David Lloyd conceived and designed the experiments, performed the experiments, analyzed the data, contributed reagents/materials/analysis tools, wrote the paper, prepared figures and/or tables, reviewed drafts of the paper.

### Animal Ethics

The following information was supplied relating to ethical approvals (i.e., approving body and any reference numbers):

All research activities were conducted in accordance with the Guidelines to the Use of Wild Birds in Research (*Fair, Paul & Jones, 2010*).

### Field Study Permissions

The following information was supplied relating to field study approvals (i.e., approving body and any reference numbers):

This research was conducted with the permission of the US Fish and Wildlife Service (Special Use Permit No. 13036), the USGS Bird Banding Laboratory (Bird-Banding Permit No. 23726 issued to JDL), and the State of Florida (Scientific Collecting Permit No. LSSC-11-00048A).

### Data Availability

Lloyd, John (2017): Mangrove Cuckoo Radio-telemetry Study. figshare. https://doi.org/10.6084/m9.figshare.4628017.v1.

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
