# Peer review of "Movements and use of space by Mangrove Cuckoos (Coccyzus minor) in Florida, USA"

_PeerJ, doi:10.7717/peerj.3534_

## Round 0.1 · original submission · Major Revisions

Dear authors

Your ms has been reviewed and the reviewers found several items which could be improved. If you are willing to revise your ms, we would be happy to review it again.

Regards

Michael Wink
Academic editor

·

Basic reporting

This is a well-written manuscript and it was a potentially interesting study because of how little we know about Mangrove Cuckoos. However, it is a manuscript high on techniques and very low on new information about Mangrove Cuckoos. For example, we don’t learn from the manuscript when birds were captured, how individuals might have differed in standard measurements or in behavior, whether or not juveniles or molting birds were captured – and when, or how these birds were dispersed relative to the density or size or habitat configuration of the mangroves or mangrove species. Did he ever locate a bird away from mangroves? If so, in what kind of habitat? Were any habitat measurements taken? He should be able to tell us something about habitat preferences in the species besides “mangroves”. We know that already. It also seems that he should have been able to learn when birds were nesting and perhaps he might have even been able to learn more about possible differences in the sexes. I suspect he has some such data and it would be worth including.

Was there ever an effort to locate birds at night in order to find where they roost?

Here are a few specific comments on the ms:

Abstract: Informative and well-written. The words “not uncommon” are a double negative that should be replaced with “common”.

Introduction:
Line 39. Delete the superfluous words “In this study”.
Line 40. Delete the word “the” before characteristics.
Line 53. “Studies” don’t have goals, people do. Delete “The goal of this study” and replace with “My goal”.
Line 59. Delete the word “planning”.
Line 75. Delete the word “because” and replace with “where”.
Field Methods
What equipment was used to broadcast? For how long? How loud? When? Possibly during nesting? Ethics of that? Whose recording? What calls? In general you’ve covered all the bases, but there are concerns about use of playback by some and it is best to address it up front.
Line 159. “Utilization” is a pompous overused word. It means nothing more than “use”. Why not tighten up by using “use”?
Results
Line 191. I’m not sure the words “By comparison” are meaningful here. It isn’t really a comparison, but a matter of relevance. I suggest deleting those words.
Line 201. Spaces are needed between some of the numbers.
Line 209. Change “not uncommon” to “common”.
Line 229. What about the possibility of mortality? There are Cooper’s Hawks and other predators that would take a cuckoo. Absence could be because of loss, with the transmitter dropped into water or otherwise incapacitated.

Literature Cited

In general there are some inconsistencies in author names. I checked several original articles and found that authors often had more than one initial in the original, but only one initial is given in the lit cited here. Others correctly have two initials.
Line 356. There is no place named “Southern Florida”. The word “Southern” here should be lowercase – it’s merely an adjective.
Line 377. I checked the BNA account for Mangrove Cuckoo and it was published in 2012, not 2010.

Experimental design

OK. But since you had to capture the birds and handle them, it would seem that you should have maximized information you could learn from them. Standard measurements? Eye and flesh colors documented?

Validity of the findings

What is reported seems fine.

Additional comments

Covered above,

Reviewer 2 ·

Basic reporting

For the most part, I think the paper is well composed and well written and the material flows sequentially and logically. The tone is professional yet relaxed and easy to understand.

I didn't see any hypotheses in the paper. It seemed to be a descriptive record of observed movement data, instead of a test of a hypothesis.

Experimental design

I was troubled that the author did not address some potentially important biases.

First, isn't there an obvious bias in censoring the individuals that are most likely to wander (lines 157-158)? The resulting data would then be biased low because birds that moved more widely were censored or were rarely located, thus leading to an inaccurate picture of how far cuckoos move.

Second, isn't there a sampling bias (i.e., non-random trapping locations) that needs to be addressed? Capturing only cuckoos that were on small public lands surrounded by development would inevitably lead to conclusions that apply only to cuckoos that live on small public lands surrounded by development. Conclusions on lines 304-306 seem to suggest that all cuckoos have large home ranges and roam across inhospitable habitat, when in fact the paper did not sample cuckoos that live in larger tracts of mangroves. No one disputes that working in remote locations inside mangrove forest would be exceptionally difficult if not impossible --- the point is that this potential bias needs to be addressed as a caveat if it cannot be directly measured. We don't know whether cuckoos in mangrove forest might have smaller ranges and be more sedentary.

Third, I think the paper needs to address the issue of battery life. It's not clear why monitoring ended when it did, and whether that was because of dead batteries or because of limited personnel. How do we know that batteries didn't just die and birds remained present in consecutive years?

Fourth, why isn't the frequency of presumed movements outside of the study area reported in more detail, discussed, modeled, or interpreted? How confident are you that such movements (lines 211-219) were captured? Can you provide some information about how often individuals were missed on a given day but then resighted on a subsequent day? Could these presence/absence data be modeled in relation to covariates such as sex or season? "Cuckoos are a wide ranging animal" seems to be an important theme of the paper, which makes the reader wonder why the movement data aren't analyzed more quantitatively.

Validity of the findings

I think some of the conclusions in the Discussion might go a bit beyond the data. I found the speculations interesting and well written, but they need to be identified as speculation.

Can you please provide a test for correlation between N locations and home range size? Scatter plot might be useful.

I did not find Figure 2 compelling. I wasn't sure what the message was. Please clarify.

The author might want to note that the data do not demonstrate any effect on nesting status or any other demographic variable, so any “effects” that are mentioned are speculative.

---

## Round 0.2 · Minor Revisions

Dear author

As you see from the comments our reviewers are not happy that you did not follow their helpful and important recommendations. Make sure that you address these questions in your next revision- the reviewers will see the revision!

Regards
Michael Wink
Academic editor

·

Basic reporting

I was frustrated to find that few of my earlier comments resulted in any change. At the very least the use of “not uncommon” (line 23) should be changed. This is a double negative – a grammatical error that means nothing more than “common”. Less serious is the continued use of “utilization” – 11 letters when 3 -- “use” -- mean exactly the same thing.

Experimental design

See above and below.

Validity of the findings

See above and below.

Additional comments

In an earlier paper (Lloyd and Doyle 2011. Journal of Field Ornithology 82(2):138,) noted that “…we do not know if Mangrove Cuckoos in Florida are migratory or resident.” Doesn’t their having confirmed that they were present on their study area year round suggest that at least some are likely resident? Why didn’t Lloyd at least provide some commentary on that prospect?

While the sexes of adult Mangrove Cuckoos are believed to be monomorphic in plumage, adults and juveniles are readily separated by plumage (see Pyle et al. 1997. Identification guide to North American birds. Part I. Columbidae to Ploceidae. Slate Creek Press, Bolinas, California). Were any of the birds fitted with a transmitter juvenile or second-year birds? There is so much more we could have learned from these birds!

Standard wing and tail measurements reported strongly overlap between the sexes, but Pyle et al. suggest that cloacal characteristics might allow ID of the sex of breeding birds (male with a more protruding cloaca). It is also possible that females could be identified by the presence of a more flaccid, larger cloacal opening resulting from egg laying.

Flesh colors of the facial region of Mangrove Cuckoos are distinctly different during the breeding season. Were these not noted? Also, there are breeding records that would provide at least some boundaries on the nesting season. Stevenson and Anderson (1994. The birdlife of Florida. University of Florida Press, Gainesville) and discuss the probable breeding season. Why isn’t that literature used and the data looked at relative to the likely breeding season? For many (most?) species, home range size is minimal during nesting, thus allowing maximum food delivery to young. Furthermore, in other species adults tend to feed farther from nest sites both before beginning nesting and after fledging of young – leaving available food for easy access when there are young in the nest. Without considering such factors, it seems that the significance of the data collected is seriously impaired. Certainly a putative breeding season could be identified and birds followed during that period compared with birds followed before and after the likely breeding season.

Finally, a factor not quantified or discussed is the size/age of the mangroves, the linear nature of the habitat in many areas (following shorelines), and the extent to which the birds were using mangrove islands as opposed to mangrove fringes of shorelines. As a matter of history, the mangroves of south Florida were decimated in the late 1800s and early 1900s by their use in the furniture industry. What we have today is regrowth that doesn’t approach their potential of 80 feet tall or more. I’ve worked in red mangrove forests in Costa Rica where the mangroves can exceed 100 feet and where a human can walk under the prop roots. The stature of the Florida habitat of the Mangrove Cuckoo may be on the rebound (where it is allowed to grow) while areal extent of the habitat continues to decline. Discussion could – and should have discussed such factors relative to data obtained and the conservation of the species.

Reviewer 2 ·

Basic reporting

Interesting study. Revised manuscript is clearer.

Experimental design

Clear statements are included about why and how methodologies were chosen.

Validity of the findings

I still find the conclusions of the paper are too broadly generalized beyond the study area. I strongly recommend that the Discussion include these types of statements: "We were only able to sample cuckoos living in small public lands surrounded by urban development. It is possible that cuckoos living in more extensive mangrove forests, which constitute the majority of mangrove cuckoo habitat, would exhibit different movement patterns and have smaller or larger home ranges."

---

## Round 0.3 · accepted · Accept

Thanks for the revisions- now your ms can be accepted

·

Basic reporting

I greatly appreciate the author's comments of rebuttal and respect his conservative approach. I disagree with his decision and rationale for using "utilization" rather than "use". Such is merely an editorial matter, not one of signficance -- except in terms of space on the printed page.

This is a useful manuscript, but I still feel that the ms should have provided more information, including identification of such questions as he raises in his rebuttal, noting the "known" timing of nesting, the need for more such data, some of the explanation as to why he didn't include more information relative to age, sex, etc., and more discussion with some of the speculation and suggestions for further work that is needed. His methods might note what measurements he did take and a table or figure summarizing those would be useful for future comparisons -- especially considering the scarcity of such information.

I nonetheless recommend publication.

Experimental design

OK

Validity of the findings

OK

Additional comments

OK